# Yellow Leaf Disease Resistance and *Melanaphis sacchari* Preference in Commercial Sugarcane Cultivars

**DOI:** 10.3390/plants12173079

**Published:** 2023-08-28

**Authors:** Luiz Eduardo Tilhaqui Bertasello, Marcel Fernando da Silva, Luciana Rossini Pinto, Paula Macedo Nóbile, Michele Carmo-Sousa, Ivan Antônio dos Anjos, Dilermando Perecin, João Roberto Spotti Lopes, Marcos Cesar Gonçalves

**Affiliations:** 1School of Agricultural and Veterinary Sciences-FCAV, São Paulo State University-UNESP, Jaboticabal 17884-900, Brazil; luiz.bertasello@unesp.br (L.E.T.B.); luciana.rossini@sp.gov.br (L.R.P.); dilermando.perecin@unesp.br (D.P.); 2Sugarcane Research Centre, Instituto Agronômico de Campinas-IAC, Ribeirão Preto 14001-970, Brazil; marcel.fernsilva@gmail.com (M.F.d.S.); paulamanobile@gmail.com (P.M.N.); ivan.anjos@sp.gov.br (I.A.d.A.); 3Department of Entomology and Acarology, Escola Superior de Agricultura Luiz de Queiroz (ESALQ), University of São Paulo, Piracicaba 13418-900, Brazil; m.sousatimossi@gmail.com (M.C.-S.); jrslopes@usp.br (J.R.S.L.); 4Crop Protection Research Centre, Instituto Biológico-IB, São Paulo 04014-002, Brazil

**Keywords:** aphid vector, sugarcane viruses, plant resistance, *Saccharum* spp.

## Abstract

Sugarcane yellow leaf disease (YLD) caused by sugarcane yellow leaf virus (ScYLV) is a major threat for the sugarcane industry worldwide, and the aphid *Melanaphis sacchari* is its main vector. Breeding programs in Brazil have provided cultivars with intermediate resistance to ScYLV, whereas the incidence of ScYLV has been underestimated partly due to the complexity of YLD symptom expression and identification. Here, we evaluated YLD symptoms in a field assay using eight sugarcane genotypes comprising six well-established commercial high-sucrose cultivars, one biomass yield cultivar, and a susceptible reference under greenhouse conditions, along with estimation of virus titer through RT-qPCR from leaf samples. Additionally, a free-choice bioassay was used to determine the number of aphids feeding on the SCYLV-infected cultivars. Most of the cultivars showed some degree of resistance to YLD, while also revealing positive RT-qPCR results for ScYLV and virus titers with non-significant correlation with YLD severity. The cultivars IACSP01-5503 and IACBIO-266 were similar in terms of aphid preference and ScYLV resistance traits, whereas the least preferred cultivar by *M. sacchari*, IACSP96-7569, showed intermediate symptoms but similar virus titer to the susceptible reference, SP71-6163. We conclude that current genetic resistance incorporated into sugarcane commercial cultivars does not effectively prevent the spread of ScYLV by its main aphid vector.

## 1. Introduction

Sugarcane (*Saccharum* spp.) is a major cash crop cultivated in tropical and subtropical regions for sugar and energy production [1,2]. Aside from ethanol production from sucrose and starch, this crop has potential for biofuel and bioelectricity production from lignocellulosic biomass and industrial residues [1,3,4]. Several diseases hamper sugarcane production [2,5], with yellow leaf disease (YLD), caused by sugarcane yellow leaf virus (ScYLV), being one of the most important and widespread viral diseases among sugarcane growing countries [6,7,8]. The disease is responsible for yield losses ranging from 10%–30% [9,10] to up to 60% [11], and has been controlled mainly by the use of resistant cultivars and certified virus-free seed cane [6,12].

ScYLV is a member of the genus *Polerovirus*, family *Solemoviridae*, and is primarily transmitted through infected vegetative cuttings [13]. Similar to other members of the genus *Polerovirus*, ScYLV is phloem-limited, with secondary transmission taking place through extensive feeding by aphids in a persistent circulative manner [14]. Among aphid species that colonize sugarcane, *Melanaphis sacchari* (Hemiptera: Aphididae) has been reported as the most prominent vector of ScYLV [15,16,17,18]. The most predominant YLD symptom is the yellowing of the midrib on the abaxial surface of the leaf midrib that can spread laterally to the leaf lamina, which afterwards becomes dry and bleached, with tissue necrosis eventually taking place, starting from the leaf tip [6,8]. Significant sucrose accumulation in leaves of cultivars with yellowing symptoms has been reported [11,19,20]. In some cultivars, YLD symptoms may also include reddish coloration of the midrib on the adaxial surface and shortening of terminal internodes [6,7,8,21]. The midrib yellowing usually appears between 6 and 8 months of crop growth, being usually favored by abiotic stresses, such as drought and cold [22,23], whereas it is not a specific response to ScYLV, but also associated with other biotic and abiotic factors [24]. Additionally, the asymptomatic phase of YLD is a major epidemiological factor contributing to the primary spread of ScYLV [21,24].

The complexity of YLD symptom expression challenges the phenotyping for the disease resistance, therefore supportive tools for ScYLV detection have been developed. These include methods that rely on ScYLV-specific polyclonal antiserum, viz., tissue blot immuno-assay (TBIA) and double antibody sandwich (DAS)-ELISA [25,26]; and on ScYLV-specific primers, viz., conventional reverse transcription–polymerase chain reaction (RT-PCR) [27], real-time quantitative PCR (RT-qPCR) [28,29], and methods based on isothermal amplification and gel-free detection of amplification products, which may consist on molecular beacons and a sequence detection system, i.e., the AmpliDet RNA system [30], or visualization of solution color change within microcentrifuge tubes, i.e., RT-loop-mediated isothermal amplification (LAMP) [31]. The serological assays, which have been widely used for ScYLV diagnosis [20,32,33,34,35], have demonstrated lower sensitivity in comparison with molecular diagnostic tools [28,30,36,37,38]. Among the latter, RT-qPCR allows precise detection and estimations of ScYLV titer via either absolute or relative quantifications, enabling the phenotyping of sugarcane clones for YLD in conditions of absence of disease symptoms [38,39].

Several studies have reported that virus-mediated changes in host plants influence vector behavior in ways that enhance virus transmission, which include alterations in host plant defenses, nutritional properties, and sensory cues presented to the vector insects [40,41]. Additionally, viral infections in many cases are also beneficial for vector colonization, which include increased vector performance and possible synergism in compromising host plant perception and defenses, thus indicating mutualistic relationships [41,42,43,44,45,46,47]. Host plant resistance to insect vectors, on the other hand, can lead to the decrease in viral disease incidence, especially for propagative-persistent and circulative-persistent virus transmission modes [48,49]. The general expression of plant resistance against insects, including vectors of viral diseases, relies on resistance mechanisms such as antixenosis, that is the non-preference of the insect due to repellence or deterrence phenomena, affecting insect settling and feeding behaviors; antibiosis, consisting of adverse effects of host plants on the biology of insects; and tolerance, which allows plants to maintain growth and production after insect feeding [50,51,52,53,54]. In sugarcane, sources of resistance against *M. sacchari* involving antibiosis and antixenosis have been identified by biological assays, while electrical penetration graph (EPG) recordings demonstrated the unsuccessful feeding of *M. sacchari* on phloem vessels in aphid-resistant cultivars [51,55,56,57].

Conventional sugarcane breeding in Brazil has provided cultivars with tolerance or intermediate resistance to ScYLV [6]; however, since most sugarcane cultivars do not express visual symptoms [6,21,38], little is known about the impact of YLD in Brazilian sugarcane areas. On the other hand, the phenotyping based on virus titration and further knowledge about plant-virus-vector interactions have been stated as promising strategies for the selection of genotypes with durable resistance to ScYLV; however, there is scant information about the virus titers in the sugarcane genotypes and their resistance to aphid infestations [38,51]. In this regard, the present study aimed to evaluate YLD symptoms, ScYLV titration, and the preference of aphid vector *M. sacchari* of eight sugarcane genotypes, comprising well-established commercial high-sucrose cultivars and a biomass yield cultivar.

## 2. Results

### 2.1. YLD Phenotyping

Variance analysis showed significant variation among genotypes in response to YLD (*p* < 0.05; Appendix A). According to the LSD test, SP71-6163 was the most susceptible over all the MAP evaluations. At 8 and 9 MAP, the genotypes IACSP95-5000 and IACSP96-7569 were the most susceptible to YLD after SP71-6163, with IACSP96-7569 not differing from IACSP95-5094, IACSP01-5503, and IACSP01-3127 at 8 MAP; and from IACSP95-5094 at 9 MAP. At 10 and 11 MAP, IACSP95-5000 was the second most susceptible to YLD, followed by IACSP95-5094, IACCTC05-2562, and IACSP96-7569 at 10 MAP; and by IACSP96-7569 at 11 MAP (Table 1). The most resistant genotype was IACBIO-266, with absence of YLD symptoms in all evaluations. The remaining genotypes showed significantly higher YLD symptom means in comparison to IACBIO-266, at 6 MAP (IACSP01-5503), at 9 MAP (IACCTC05-2562), and at 11 MAP (IACSP95-5094, IACCTC05-2562, IACSP01-5503, and IACSP01-3127).

### 2.2. RT-qPCR and Virus Titer

The raw Ct values of the six candidate reference genes and the target gene, i.e., 181 bp fragment of ScYLV capsid protein, along with their mean Ct values, coefficient of variation (CV%) values, individual and mean amplification efficiency, and correlation coefficients (R^2^) are listed in the Appendix A. The mean amplification efficiency among these genes ranged from 1.85 ± 0.09 to 2.13 ± 0.14, which is close to the optimal range of 90–110% [58]. The mean Ct values of the target gene per genotype were 27.23 ± 1.35, 28.47 ± 1.75, 30.22 ± 0.6, 30.84 ± 3.69, 32.33 ± 2.49, 34.23 ± 6.16, 34.38 ± 4.11, and 36.49 ± 0.85 for SP71-6163, IACSP01-3127, IACSP95-5094, IACSP95-5000, IACSP96-7569, IACBIO-266, IACCTC05-2562, and IACSP01-5503, respectively (Appendix A). Melting curve analysis showed unique peaks of fluorescence for all primer pairs, indicating RT-qPCR amplification of single fragments (Appendix A). Positive RT-qPCR results for ScYLV were observed for all varieties in at least three sampling times and one technical replicate (Appendix A). The stability ranking of the six candidate reference genes according to the NormFinder algorithm was the following: UBC18 > TUB > GAPDH > TPI > UK > SAND (Appendix A). Thus, UBC18 was employed as the reference gene for the virus titer assessment and considered as a prior in the Bayesian MCMC model for the transcript abundance analysis.

Virus titer variations significant at the 95% credible interval among sampling times were observed for all genotypes, except for IACSP01-5503 and SP71-6163 (Figure 1). These variations comprise the increase in virus titer between 0 and 8 MAP, decrease between 8 and 10 MAP, and increase between 10 and 12 MAP for IACCTC05-2562 and the reverse trend, i.e., the decrease between 0 and 8 MAP, increase between 8 and 10 MAP, and decrease between 10 and 12 MAP, for IACBIO-266. Increases in virus titer between 0 and 8 MAP were also observed for IACSP95-5000 and IACSP96-7569, with the latter also showing an increase in virus titer between 8 and 12 MAP, while decreases between 0 and 12 MAP were observed for IACSP01-3127 and IACSP95-5094, and also between 8 and 12 MAP for the latter. The genotypes with significantly higher virus titer across sampling times were SP71-6163, IACSP01-3127, and IACBIO-266 at 0 MAP; SP71-6163, IACSP95-5000, and IACSP95-5094 at 8 MAP; SP71-6163, IACSP01-3127, and IACSP95-5000 at 10 MAP; and IACSP95-5000, IACSP96-7569, SP71-6163, IACCTC05-2562, IACSP01-3127, and IACSP95-5094 at 12 MAP. It is noteworthy that the susceptible reference genotype, i.e., SP71-6163, was amongst those with higher virus titer across all sampling timepoints. The virus titer fold change in relation to SP71-6163 revealed either significantly lower SCYLV titers or non-significant values in the remaining cultivars across sampling timepoints, according to the 95% credible intervals (Figure 2). The average values of the significant virus titer fold decrease ranged from 4.48 × 10^−2^ to 2.7 × 10^−4^ for IACSP95-5094, IACSP96-7569, IACSP01-5503, IACSP95-5000, and IACCTC05-2562 at 0 MAP; from 2.91 × 10^−1^ to 9.11 × 10^−9^ for IACSP95-5094, IACSP96-7569, IACCTC05-2562, and IACBIO-266 at 8 MAP; from 3.15 × 10^−1^ to 1.21 × 10^−11^ for IACSP95-5000, IACBIO-266, IACSP95-5094, IACSP96-7569, IACSP01-5503, and IACCTC05-2562 at 10 MAP; and from 9.93 × 10^−4^ to 1.7 × 10^−6^ for IACSP01-5503 and IACBIO-266, respectively. The non-significant fold changes were observed for IACSP01-3127 and IACBIO-266 at 0 MAP; IACSP95-5000 at 8 MAP; IACSP01-3127 at 10 MAP; and IACSP95-5000, IACSP96-7569, IACSP01-3127, IACCTC05-2562, and IACSP95-5094 at 12 MAP.

Virus titer was not correlated with YLD symptom expression among sugarcane cultivars (r^2^ = 0.3254328; *p* = 0.1394). An evident contribution to the non-significative correlation between these traits was the significant virus titer fluctuation in the resistant cultivar, IACBIO-266. Other contributions for the poor correlation consisted of the mismatch between YLD severity and virus titer fluctuations between 8 and 10 MAP and between 10 and 12 MAP for IACCTC05-2562, and by the non-significant increases in virus titers among cultivars with increasing YLD severity across sampling timepoints, namely, IACSP01-3127, IACSP01-5503, and IACSP95-5094. On the other hand, the increasing YLD severity between 0 and 8 MAP observed for IACSP95-5000, IACSP96-7569, and IACCTC05-2562 was coherent with their increasing ScYLV titers.

### 2.3. Aphid Preference

The one-way ANOVA for each timepoint of the free-choice bioassay revealed non-significant differences among cultivars for number of *M. sacchari* (*p* > 0.05) (Appendix A). The mean number of aphids across timepoints indicated increases from 0.5 to 3 h for the cultivars IACSP01-5503, IACSP96-7569, IACBIO-266, IACSP01-3127, and IACSP95-5000; decreases from 3 to 24 h for IACSP01-5503, IACSP96-7569, and IACSP95-5000; and increases from 3 to 24 h for IACSP01-3127. Another noteworthy change was the increase in the mean number of aphids from 6 to 24 h observed for IACCTC05-2562 (Figure 3A). Nonetheless, the two-way ANOVA with all 640 observations of the free-choice bioassay showed non-significant effects for time after release and its interaction with cultivar, while a significant F value (*p* < 0.05) was observed for the variable cultivar (Table 2). The least *M. sacchari* preferred cultivar according to the LSD test (*p* < 0.05) was IACSP96-7569, followed by IACBIO-266, IACSP01-5503, and IACSP95-5094. The latter three genotypes, on the other hand, did not differentiate from the most preferred ones, namely, SP71-6163, IACSP95-5000, IACSP01-3127, and IACCTC05-2562 (Figure 3B).

## 3. Discussion

ScYLV appeared in the mid 1990’s and became endemic in the main sugarcane-growing areas of Brazil [59], impelling the breeding for YLD resistance for over the last two decades by Brazilian sugarcane breeding programs. Such efforts have relied on phenotypical clone selection and resulted in tolerance or intermediate resistance of most of the sugarcane genotypes cultivated in the country [6]. On the other hand, the incidence of ScYLV has been underestimated, with the impacts on yield being neglected due to the near to absence of YLD symptoms in most cultivars [6]. The majority of the sugarcane cultivars assayed in the present study showed variable degrees of resistance to YLD while revealing positive RT-qPCR results for ScYLV presence, indicating that they may represent inoculum sources for the secondary spread of the virus in the field by aphid vectors. Although Ct values for IACBIO-266 were also recorded by Burbano et al. [38], ScYLV was considered non-detectable in this cultivar due to melting temperatures below 81.5 °C in two out of three blocks from a field experiment under a randomized complete block design. Here, melting temperatures above this threshold were observed for IACBIO-266 in all cDNA samples, each representing pools of RNA from three biological replicates, i.e., blocks, and thus were considered positive for ScYLV.

The virus titer assessment of the cultivars assayed here revealed significant fold decreases ranging from 3.15 × 10^−1^ to 1.21 × 10^−11^ in relation to the susceptible reference genotype, while ScYLV titers also varied among sampling timepoints for most of the cultivars and showed non-significant correlation with YLD severity. Previous studies have investigated the correlation between ScYLV titer and YLD symptom expression in sugarcane genotypes. The presence of ScYLV, revealed by TBIA, and the development of YLD were reported as correlated in a non-strict manner by Lehrer and Komor [20] based on the higher YLD incidence and severity within ScYLV-infected cultivars. According to the authors, the inconsistencies were due to the differential responses of some cultivars to ScYLV infection and virus titer fluctuations with the plant age, above and below the threshold detection of TBIA. Such titer fluctuations were confirmed by RT-qPCR, in a range of 10^3^–10^4^-fold, with the causes remaining unknown [60]. RT-qPCR assays have also revealed that ScYLV titers in resistant sugarcane cultivars were at least 10^6^-fold lower than in susceptible ones [60], while symptomatic sugarcane plants showed 10^3^–10^4^ and virus titers up to 291-fold higher than the asymptomatic ones, with evidence pointing to a ScYLV titer threshold to cause YLD symptoms [60,61]. Accordingly, such virus titer thresholds appear to vary among sugarcane genotypes and may also vary with environmental conditions [21,38,60]. In assays for resistance screening to ScYLV on sugarcane panels comprising basic germplasm, elite clones, and commercial cultivars, no quantitative correlation between YLD severity and ScYLV titer was observed by Pimenta et al. [62], while Burbano et al. [38] reported high coefficient of correlation values for the moderately susceptible and susceptible classes of genotypes and very low and non-significant values for the moderately resistant ones.

The absence of YLD symptoms in all of the IACBIO-266 phenotyping assays, in combination with the fluctuating ScYLV titers, indicates that this cultivar is able to restrict the expression of YLD symptoms even at high virus titers, what was also observed for more susceptible cultivars at 0 and 10 MAP. The resistance of this genotype found in this study may be attributed to the genetic background inherited from the female parent SES069, a *S. spontaneum* accession. The contribution of *S. spontaneum* to YLD resistance has been previoulsy highlighted in studies screening sugarcane panels for resistance to ScYLV [38,63]. Conversely, aside from the contribution to the primary spread of ScYLV, during the asymptomatic phase of YLD yield losses of 20–30% have been reported [22]. Thus, the presence of ScYLV over the plant development of some genotypes requires attention, and stresses the importance of combining YLD phenotyping with ScYLV titer assessment to improve and speed up the selection of resistant genotypes [21,38].

The results obtained in the free-choice bioassay revealed IACSP96-7569 as the least preferred cultivar, followed by IACSP95-5094, IACSP01-5503, and IACBIO266, which supports the expression of antixenosis among the cultivars evaluated here. Additionally, the apparent movement of aphids among cultivars throughout the time intervals of 0.5, 1, 3, 6, and 24 h after release further indicates that host factors may have affected the settling behavior of the non-viruliferous aphids. Such outcome differs from the reported by Akbar et al. [64], in which antixenosis assays revealed no preference of *M. sacchari* and *Sipha flava* among sugarcane cultivars, with aphids finding their hosts within an hour followed by no apparent movement among cultivars for the 24 h duration of the assay. Conversely, significant differences on aphid settling on potted sugarcane plantlets were observed by Fartek et al. [51] when comparing the cultivars R 365 and MQ 76/53 under no-choice conditions at 8, 24, 48, and 72 h after aphid deposit, indicating the expression of antixenosis in the former cultivar from 8 to 24 h, followed by a similar decrease in aphid populations between cultivars from 24 to 72 h. Previous studies assessing the resistance of sugarcane to aphids have also revealed significant variance among cultivars in terms of aphid incidence [65,66] as well as life history parameters and feeding behavior [51,56,57,64,67]. In the cultivar IACSP96-7569, *M. sacchari* has also showed the worst biological performance, indicating that antibiosis is also expressed here [57]. Regarding the cultivars which showed intermediary antixenosis in this study, i.e., IACSP95-5094, IACSP01-5503, and IACBIO266, data of resistance to non-viruliferous *M. sacchari* in terms of reproductive parameters were correspondingly observed for the first two cultivars, whereas the latter was amongst the most favorable to *M. sacchari* reproduction [57], exhibiting diversity in terms of vector resistance mechanisms among these cultivars.

The host-vector interaction is another determining factor in viral epidemiology, with the effects of vector resistance on disease spread being strongly dependent on the virus transmission modes [48]. For instance, increase in the spread of non-persistently transmitted viruses has been reported in aphid-resistant cowpea plants [68], while reduction in the spread of persistently and semipersistently transmitted viruses associated with host plant resistance to whitefly has been reported in tomato [49,69]. Regarding sugarcane YLD, modest and positive correlations between aphid and ScYLV incidence were reported by Fartek et al. [66], indicating some degree of independence between these traits, possibly due to genotypes possessing virus resistance genes while showing susceptibility to *M. sacchari*. Viswanathan et al. [70] investigated the population dynamics of *M. sacchari* in field assays comprising hybrid varieties, inter-specific hybrids, and sugarcane varieties under typical tropical conditions in India, and reported higher aphid populations in plants expressing YLD comparatively to the symptomless ones from the same genotype. Additionally, the authors also observed comparatively higher number of aphids on the YLD-susceptible varieties, whereas instances of genotypes with high YLD severity and low aphid populations indicated that a limited population or even a single viruliferous aphid may spread the disease. Here, the cultivars IACSP01-5503 and IACBIO266 have shown agreements between aphid antixenosis and virus resistance traits. More specifically, the cultivar IACSP01-5503 showed intermediary resistance to YLD, relatively lower ScYLV titer over all evaluations, and lower *M. sacchari* preference. Similarly, IACBIO266 showed high resistance to YLD whilst relatively less preferred by the aphid vector. On the other hand, the putative resistance mechanisms carried by IACSP96-7569 involving antibiosis and antixenosis seemed to not prevent the secondary infection by ScYLV, which may have occurred either under our experimental conditions or previously in the field, followed by spread via virus-infected seed cane.

The present results indicate that current genetic resistance incorporated into sugarcane commercial cultivars may restrict YLD incidence in the field, but not necessarily reduce ScYLV titer. Further studies are necessary to determine the extent of varietal degeneration caused by YLD, unveil which genes are behind the genetic resistance of sugarcane against aphid vectors, and establish the relationship of this trait with the spread of ScYLV.

## 4. Material and Methods

### 4.1. Plant Material

YLD was evaluated in eight sugarcane cultivars, namely IACSP95-5094, IACCTC05-2562, IACSP01-5503, IACSP96-7569, IACSP01-3127, IACSP95-5000, SP71-6163, and IACBIO-266, maintained at the IAC Sugarcane Breeding Station, Ribeirão Preto, São Paulo, Brazil (latitude 21°10′39″ S and longitude 47°48′37″ W). The first six consist of high-sucrose cultivars currently cultivated in Brazil, while SP71-6163 was replaced from commercial sugarcane areas due to its high susceptibility to YLD, and therefore adopted as the susceptible reference in this study. IACBIO-266, in turn, is an energy cane cultivar aiming for biofuel production. Buds were obtained from stalks collected from sugarcane stools and planted in plastic trays containing commercial substrate composed by sphagnum turf, expanded vermiculite, dolomitic limestone, and gypsum. The trays were placed in a photoperiod chamber at 28 °C and photoperiod of 12 h for sprouting/germination. The obtained seedlings were individually transferred to stiff plastic tubes (volume of 50 cm^3^) and maintained on stands in an aphid-proof greenhouse during ninety days until being taken to the field.

### 4.2. Aphid Rearing

A colony of *M. sacchari* was established from a clone provided by Centre of Molecular Biology and Genetic Engineering, Campinas, São Paulo, Brazil (latitude 22°49′13″ S and longitude 47°3′34″ W), which colonies originated from aphid samplings previously performed at the IAC Sugarcane Breeding Station. Viruliferous aphid colonies were reared on detached leaves of RT-PCR tested ScYLV-infected plants of the cultivar IACSP95-5000 [57]. The youngest fully emerged leaves of these plants were cut, and the fragments were kept in test tubes containing 1% agar solution within a growth chamber (12 L:12 D, 29 ± 1 °C). Non-viruliferous aphid colonies, on the other hand, were reared on fragments of the youngest fully emerged leaves of healthy plants of IACISP95-5000 obtained from meristem tip culture, maintained under aphid-proof greenhouse conditions, and indexed by RT-PCR. In both conditions the sugarcane leaves were changed every 6 ± 1 day and newly born individuals were sampled and reared on new detached leaves. The presence and absence of the virus in the aphid colonies were confirmed by RT-PCR analysis.

### 4.3. YLD Phenotyping

Viruliferous *M. sacchari* colonies were released biweekly in the aphid-proof greenhouse during ninety days to promote uniform ScYLV infection. After the last release of viruliferous aphid colonies, sugarcane plants were taken to the field (November 2019) in a complete randomized block design with six replicates. Each plot consisted of four lines spaced 1.5 m between rows and 0.5 m between plants. Six additional biweekly releases of viruliferous aphids were performed in the field, after planting. YLD symptoms were recorded in six monthly evaluations in the two inner lines of each genotype plot, while the outer lines served as side borders, starting six months after planting (MAP; May 2020) up to eleven MAP (October 2020). A diagrammatic symptoms scale was used to evaluate YLD severity: (1) green leaf without symptoms; (2) slight yellowing of the midrib and leaf blade; (3) intense yellowing of the midrib and partial yellowing of the leaf blade; (4) intense yellowing of the midrib and leaf blade (Appendix A). The symptom records were performed by four different evaluators to reduce the experimental error. Data were ln (x + 5) transformed and the normality was tested with Shapiro–Wilk W test. The transformed data were submitted to variance analysis using the general linear model (Proc GLM) and mixed model (Proc MIXED) procedures of SAS package version 9.3 [71]. Means were subsequently compared using the least significant difference test (LSD test) at *p* < 0.05.

### 4.4. RNA Extraction and RT-qPCR

ScYLV diagnosis and estimation of virus titer were performed via RT-qPCR from field samples collected at 0, 8, 10, and 11 MAP. For each cultivar x MAP, total RNA was extracted from three biological replicates, corresponding to the first leaf with visible dewlap from top to bottom (+1 leaf), and stored at −80 °C. Total RNA was extracted using 1000 µL of Trizol (Invitrogen, Carlsbad, CA, USA) per 100 mg of leaf tissue, according to the manufacturer’s instructions. RNA concentration was evaluated using a NanoDrop (Termofisher Scientific, Wilmington DE, USA). RNA integrity was checked by electrophoresis in a 1% agarose gel stained with ethidium bromide in TAE buffer. The RNA from the three biological replicates were pooled together for each cultivar x MAP for cDNA synthesis (n = 32 RNA pools). From these pools, 1 µg of RNA was treated with RQ1 RNase-Free DNase to remove genomic DNA, following manufacturer’s instructions (Promega, Fitchburg WI, USA). The reverse transcription was performed using the GoScript Reverse Transcription System (Promega), in a final volume of 10 µL containing 500 ng of DNAse treated RNA, 2 µL of GoScript Reaction Buffer + Random Primers, and 1 µL of GoScript Enzime Mix. The RT-qPCR reactions were performed on a IQTM5 Real-Time PCR Detection Systems BIO-RAD. The reaction mixture consisted of 3 μL of (1:10) diluted cDNA, 5 μL of Syber qPCR, and 2 μM of each forward and reverse primers in a total volume of 10 μL. The thermal cycling conditions consisted of denaturation at 95 °C for 2 min, followed by 40 cycles at 95 °C for 15 s, 60 °C for 1 min, 95 °C for 1 min, and 55 °C for 1 min. The target amplicon was 181 bp long, obtained with forward (5′-GGACCGAACCTATCTCAGTAC-3′) and reverse (5′-TAGTAATCTTGGAGCCTGTTGTTG-3′) primers annealing to the ScYLV capsid protein [18]. Six potential reference genes previously reported in sugarcane, i.e., triosephosphate isomerase (TPI), Uridylate kinase (UK), SAND protein family (SAND), Ubiquitin-conjugating enzyme 18 gene (UBC18), glyceraldehyde-3-phosphate dehydrogenase (GAPDH), and TUBULIN (TUB) (Appendix A), were tested by RT-qPCR in another set of four cDNA pools, each representing a sampling time and comprising the eight genotypes, using three technical replicates plus three non-template controls (NTCs), yielding a total of 90 RT-qPCR reactions. The RT-qPCR efficiency of each primer set and the corresponding product threshold cycle (Ct) were obtained via LinReg PCR analysis [72]. The Ct and reaction efficiency data from the potential reference genes were used as input in NormFinder software [73] for stability evaluation. The target gene and the most stable gene, according to NormFinder, were amplified in the 32 cDNA pools using three technical replicates, yielding a total of 192 RT-qPCR reactions plus three NTCs per gene per RT-qPCR plate. The amplicon specificity was assessed for all primer sets via dissociation curve profiles (melting curves). Regarding the target gene, cDNA samples with melting temperatures of 82.5 ± 1.0 °C were considered positive for ScYLV [74]. RT-qPCR data were analyzed using the Poisson-lognormal generalized mixed model and the Markov Chain Monte Carlo (MCMC) sampling scheme from MCMC.qpcr package [75] using genotype and time point as fixed factors, with the best reference gene added as prior to the model function, allowing 1.2-fold changes. Posterior mean estimates of virus titer were plotted as log2 mRNA abundance with error representing 95% credible intervals, where differences among cDNA samples were considered significant when credible intervals did not overlap. Pearson correlation test between virus titer and YLD symptoms was performed using the cor.test function of the R program [76].

### 4.5. Aphid Settling Behavior

The settling behavior of non-viruliferous *M. sacchari* was investigated in a free-choice bioassay aiming to determine the aphid preference among eight ScYLV-infected sugarcane cultivars. The experiments were conducted in glasshouse facilities at Department of Entomology and Acarology, ESALQ-USP, and cubic arenas (30 × 30 × 30 cm) (Appendix A) were used to assess the preference of aphids (a similar experiment was described by Ramos et al. [77]). The cage structure was made from acrylic to allow natural light to penetrate the cage. The plants of each treatment had their leaves exposed through one of the side openings, one leaf from each plant per opening, with the treatments distributed equidistantly and interspersed. A circular acrylic arena coated with black adhesive tape was used to release the apterous aphids, and a squared black platform was fixed on the roof of the cage to support a leaf of each sugarcane cultivar (without detaching from the plant), interspersed at random and without touching (Appendix A).

Fifty apterous adult non-viruliferous *M. sacchari* were released in the circular arena after a fasting period of one hour. The number of aphids remaining on each sugarcane cultivar was counted at short (0.5, 1, 3, 6 h) and long term (24 h) intervals after release, and the assay replicated sixteen times. Each arena being a repetition, in each repetition the plants and aphids were replaced. Data were ln (x + 5) transformed and mean comparisons were performed by ANOVA with significance level at *p* < 0.05 using the SAS package version 9.3 [71].

## Figures and Tables

**Figure 1 plants-12-03079-f001:**
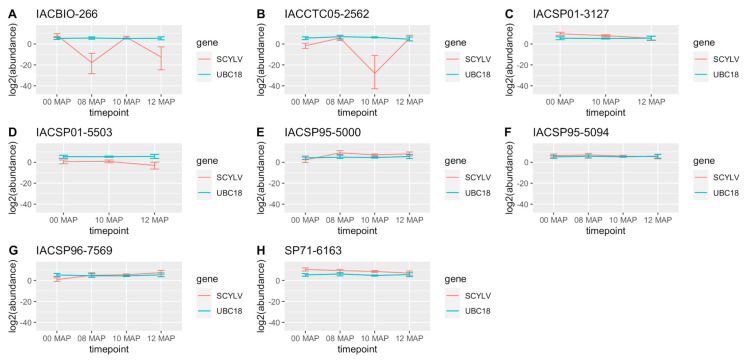
Sugarcane yellow leaf virus (SCYLV) titer (log2 abundance) in eight sugarcane genotypes across four sampling times. Data were modeled using the Bayesian MCMC sampling scheme of the ‘MCMC.qpcr’ package, considering UBC18 as a prior. Error bars represent 95% credible intervals surrounding the posterior mean estimate, with differences among samples significant at no overlapping intervals.

**Figure 2 plants-12-03079-f002:**
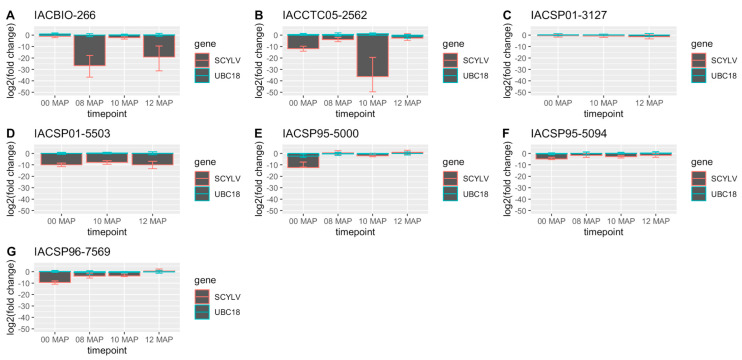
Sugarcane yellow leaf virus (SCYLV) log2 fold changes in seven sugarcane cultivars relative to the susceptible reference cultivar, SP71-6163. Data were modeled using the Bayesian MCMC sampling scheme of the ‘MCMC.qpcr’ package, considering UBC18 as a prior. Error bars represent 95% credible intervals surrounding the posterior mean estimate, with differences with SP71-6163 significant at no overlapping with zero.

**Figure 3 plants-12-03079-f003:**
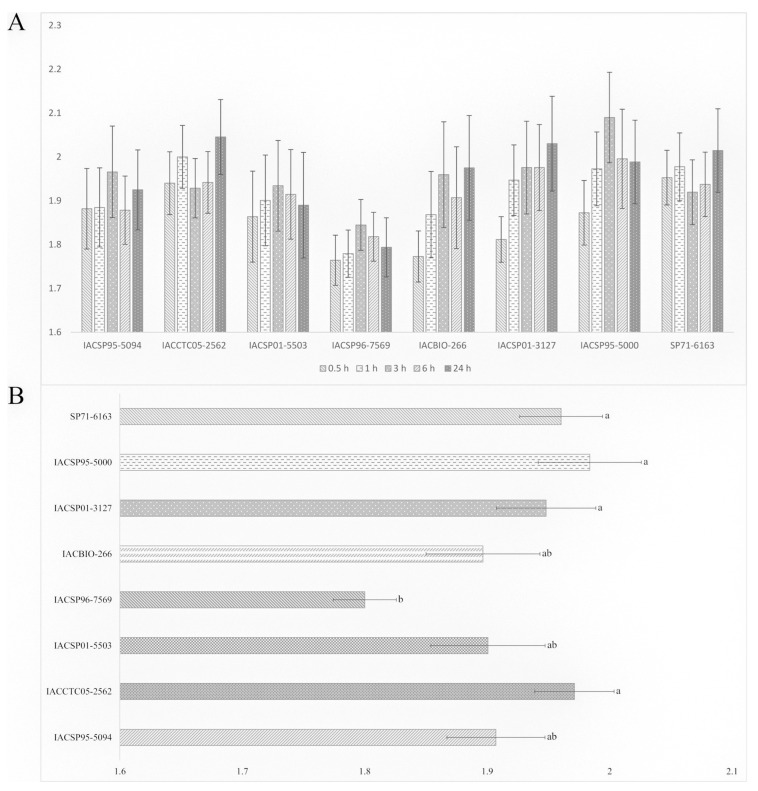
(**A**) Ln (x + 5) transformed mean number of apterous adult non-viruliferous *Melanaphis sacchari* per sugarcane cultivar at short (0.5, 1, 3, 6 h) and long-term (24 h) intervals after the release of fifty individuals in a free-choice bioassay. The differences among cultivars are not significantly different according to the LSD test after one-way ANOVA for each timepoint (*p* > 0.05). Error bars represent the standard error (n = 16). (**B**) Ln (x + 5) transformed mean number of adult non-viruliferous *M. sacchari* per sugarcane cultivar, each consisting of the mean of the five timepoints. Means with the same letter are not significantly different according to the LSD test after two-way ANOVA with all observations (*p* < 0.05). Error bars represent the standard error (n = 80).

**Table 1 plants-12-03079-t001:** Yellow leaf disease (YLD) mean severity compared by LSD test (*p* = 0.05) after ln(x + 5) transformation.

Genotypes	Symptom Evaluations							
	6 MAP	7 MAP	8 MAP	9 MAP	10 MAP	11 MAP	Mean	Class
IACSP95-5094	1.07 c	1.11 b	1.28 cd	1.78 cd	1.84 c	1.79 d	1.48 d	MR
IACCTC05-2562	1.00 c	1.00 b	1.03 d	1.43 de	1.94 c	1.78 d	1.36 d	MR
IACSP01-5503	1.21 b	1.17 b	1.17 cd	1.08 ef	1.25 d	1.35 e	1.20 e	MR
IACSP96-7569	1.00 c	1.15 b	1.53 bc	2.00 bc	1.93 c	2.1 c	1.62 c	MR
IACBIO-266	1.00 c	1.00 b	1.00 d	1.00 f	1.00 d	1.00 f	1.00 f	R
IACSP01-3127	1.00 c	1.00 b	1.10 cd	1.13 ef	1.18 d	1.38 e	1.13 e	MR
IACSP95-5000	1.00 c	1.00 b	1.78 b	2.26 b	2.93 b	3.12 b	2.01 b	MS
SP71-6163	1.98 a	2.46 a	3.29 a	3.98 a	4.00 a	4.00 a	3.29 a	S

Means with the same letter are not significantly different.

**Table 2 plants-12-03079-t002:** Two-way variance analysis (ANOVA) for the number apterous adult non-viruliferous *Melanaphis sacchari* across eight sugarcane cultivars in a free-choice bioassay.

Source	DF	Sum of Squares	Mean Square	F Value	Pr > F
Model	39	3.64844257	0.09354981	0.74	0.8748
Error	600	75.59045779	0.12598410		
Source	DF	Type III SS	Mean Square	F Value	Pr > F
Cultivar	7	1.96570427	0.28081490	2.23	0.0305
Time after release	4	0.81747984	0.20436996	1.62	0.1670
Cultivar x time	28	0.86525846	0.03090209	0.25	1.0000
CV (%)			18.47719		

## Data Availability

The data presented in this study are available in the Appendix A.

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
