# Peer review of "Yellow Leaf Disease Resistance and Melanaphis sacchari Preference in Commercial Sugarcane Cultivars"

_plants, 2023, doi:10.3390/plants12173079_

Round 1
Reviewer 1 Report
Results presented at manuscript verified importance of usage not only molecular tools at virus diagnostics, but also evaluation of YLD symptoms. Presented results gave background for further research at this field of study. I recommend include more samples (more cultivars or samples from different parts of Brasil) for future extensive study.
I propose some changes at manuscript:
Change the order of the chapters. Behind section 1. Introduction put Material and methods, then Results and Discussion. Present order of chapters at manuscript is illogical.
At section Results Figures 1 and 2 are less understandable and transparent thanks to various Y scale (log 2). I understand that there are high differences at gene content. However, I propose more unification of the scale, for example from 10 to - 40 by 5 degrees for Figure 1 and from 1 to - 45 by 5 degrees for Figure 2. Differences among individual genotypes will be better comparable.
At figure 3 it seems me not necessary to colour each column with different graphic. I propose unify them in one colour or graphic.
Reviewer 2 Report
Dear Authors,
I read carefully your submitted article Plants-2498723, and I consider it a nice and quite interesting contribution to the knowledge of sugarcane resistance to YLD through the virus transmission due to the aphid species Melanaphis sacchari . The subject of this study is important for the economy of South American Countries, such as Brazil, so I think that your article can have a wider readship in attracting not only basic researchers like plant genetists, breeders and pathologists but also agronomists and growers. The paper is well presented in the stucture , comprising an Abstract, Introduction , Results. Discussion , M & M and a references list. Also, the written language is a fine English , understandable by readers from different Countries. All the chapters satisfy and fit the main guidelines of the Journal and its scopes. However, I would like to suggest to consider a few notes put in my review of the mns (see, please, the attached word file below). The notes come from the lack of some details in M &M section , at sub-paragrahs 4.1 Plant material, and 4.2 Aphid rearing. I consider the details required as important for a better understanding of the work plan used.I also suggest to omit The supplementary Materials , from 407 to 430 lines. It can be add to the main MDPI link recorded.
Sincerely

Reviewer 3 Report
Not being an expert in molecular methods I will abstain from commenting on the methodologies used and the results obtained on this subject
Introduction/Objectives:
It is not clear what the objective of this work is and how the assays carried out may respond to this objective. The goal should be clearly indicated. This weakness is revealed also at the discussion since the results obtained are not adequately discussed following the determined objectives.
Results
Line 104 – The way in which the results of this assay are exposed may lead to inaccuracies. For example, we can't really say that in 8 MAP, IACSP96-7569 is actually different from IACSP95-5000, IACSP01-3127, IACSP01-5503 or IACSP95-5094. In fact, it is only in 11 MAP that the results are more real. I think it would be much more interesting if these results were exposed and perhaps represented in a temporal way.
Line 183 – I think the results of this assay are very incomplete, not only those that are presented in the figures, but also those that are not presented. If there is a study over time, why nothing is referred about this or represented in the text. This is a behavioural study, nothing is said about the behaviour of this aphid in the apparatus studied. On the other hand, the results of the PCA I think are unnecessary (see methodology)
Discussion
Line 283 – The discussion about the correlation between the results obtained in the 3 assays is very confusing. The results obtained in this work are discussed by mixing the discussion of the author results with the results of other studies. It must be some exposition method for the discussion to become coherent. Especially because there it seem to be differences between the results obtained in this study in relation to other studies
First, it would be important to know what the results of this study suggest in terms of what is studied in the three assays; then compare with other studies conducted by other authors, then discuss the various factors that interfere in the resistance and virus propagation; finally, their implications.
4. Material and methods
Line 393 Aphid settling behavior: It is important to define in this work what this behavior consists. This definition varies according to various authors (Ann Appl Biol 165 (2014) 3–26).
Line 399 The apparatus used to study aphid preference needs to be better explained, especially if it is not explained in previous articles (I don't see any references). Even with the figure, questions continue to arise: what is the size of the device? How big is the arena? How is the arena constituted (it is not clear what is seen in the image), does it have edges? What is represented in the image (it looks like a card with holes)? what's its goal? How are the leaves inserted into the box? How do you ensure that the leaves are at the same distance from the centre of the arena (in the figure each leaf appears to be at a different distance)? How is the influence of light controlled in this study? The distribution of the plants is random, but how are the various repetitions organized? Are the leaves of the plants replaced between each repetition? Is the order of plants changed between repetitions? Is the order of plants always constant? Behavioural studies are subject to many factors that require the most appropriate possible control. Depending on the answers to all these questions, this behavioral study may have more or less strength.
Line 403 I do not think that principal component analysis (PCA) is the appropriate way to analyse these data (there are only two variables and their effects are well determined in the analysis of variance). What's new about the PCA? Are the results of the analysis of variance compatible with the results of the dendrogram, since they are based on the same data and have the same objective.
However, if the authors consider that the PCA is an adequate analysis the justification for its use should be adequately explained. Likewise, the results of this analysis as well as the respective discussion should be more clear in the text. Otherwise, it is preferable not to use these analyses.
Suggestion: it would be important and interesting, since observations have been made over time, to show how the density of aphids evolved in each plant; Is it a quick or time-consuming choice?
In conclusion, I think it is an article that presents valid results (in the case of aphid preference depending on the validity of the methodology), but that needs important work in structural and text adjusting.
Round 2
Reviewer 3 Report
Attached file

Author Response
We thank the reviewer for the comments and suggestions to improve our manuscript (plants-2498723). The suggestions were accepted and incorporated to the text. These parts are highlighted in yellow in the text. Responses to reviewer’s comments are below.
COMMENTS TO THE AUTHOR:
Reviewer #3:
1) The way multivariate analysis is explained is not credible. A table (Table 3) is added that does not bring anything new to the figure. The representation of the figure is not typical of a PCA, it seems more the result of a correspondence analysis: what do the points with the cultivar designations represent? In terms of discussion, beyond the statistical significance, what is the biological significance of these results (PCA and dendrogram)? Nothing is indicated in the discussion.
Response: The addition of Table 3 describing factor loadings aimed to better illustrate the variables used for the PCA, i.e., time after release. However, we agree that such an addition did not improve the presentation of the free-choice bioassay results. The points shown in figure 4 of the previous version of the manuscript represent the mean aphid number for each sugarcane cultivar, while the arrows were the eigenvectors drawn by passing loadings = TRUE in the prcomp function of the R program. In the current revised version of the manuscript, PCA was removed and therefore not addressed in the discussion section, since we agree the exploratory multivariate analysis did not properly demonstrate the behavior of M. sacchari in the free-choice bioassay. Instead, ANOVA was further explored and discussed. Please, see query#4 addressed below.
2) The PCA is mainly used to understand which variables are related. It would be expected that all variables would be positively correlated because in reality it is a single variable that only varies over time and this is easily verifiable by the Supplementary figure S5 graph.
Line 329 -According to the biplot graph, IACSP95-5000 was discriminated with the highest settling of M. sacchari at 3 and 6 h, while IACSP96-7569 was discriminated as the least preferred cultivar. In terms of biology/behavior what does this means?
Response: According to the literature consulted, PCA can be used on a wide range of data with no need for rigorous assumptions when the main objective is descriptive, which includes time series data [1,2]. However, for inferential purposes, the non-independence of variables indeed requires caution.
The highest settling of M. sacchari on IACSP95-5000 at 3 and 6 h after release reveals differences in aphid settling on this cultivar across timepoints, which indicates movement of aphids during the evaluation of the free-choice bioassay. The settling behavior of M. sacchari over time was addressed in the revised version of the results and discussion sections, as described in the response to query#4 below.
- Jolliffe IT, Cadima J. 2016 Principal component analysis: a review and recent developments. Phil. Trans. R. Soc. A 374:20150202. http://dx.doi.org/10.1098/rsta.2015.0202
- Jolliffe IT. 2002 Principal component analysis, 2nd edn. New York, NY: Springer-Verlag.
3) For the graph with the comparison of the averages of the number of aphids in the various cultivars, what period of observation time? It should be mentioned in the caption and in the text.
Response: The values shown for each cultivar in figure 3 are the means of 16 replicates and five timepoints. The following change was performed in manuscript:
Lines 329-332: “B) Ln (x+5) transformed mean number of adult non-viruliferous M. sacchari per sugarcane cultivar, each consisting of the mean of the five timepoints. Means with the same letter are not significantly different according to the LSD test after two-way ANOVA with all observations (P < 0.05). Error bars represent standard error (n = 80).”
4) The graph 5 presented in the supplementary material indicates that the results over time for the number of aphids should be better explained. What justifies the averages decreasing over time? Counting errors, death of aphids, or movement of aphids out of the plant? This graph is much more interesting than the graphs of the multivariate analyses, because it is in fact this graph that most accurately represents the result of the behavioural study.
I think these issues should be reflected in a definitive version of this article.
Response: The authors agree about the need for a better explanation for the changes in aphid settling over time. The changes over time in the mean aphid numbers are likely due to their movement among cultivars since the total number of aphids on cultivars per assay replication tended to increase with time, with exceptions for replications 11 and 12 (data not shown). The following changes were made in the manuscript:
Removal of the PCA analysis from the Methods section in line 220.
Addition of a supplementary table with one-way ANOVA results for each of timepoints after aphid release: “Supplementary Table S7. One-way variance analysis (ANOVA) for the number apterous adult non-viruliferous Melanaphis sacchari for the timepoints of 0.5, 1, 3, 6, and 24 h after release, in a free-choice bioassay with eight sugarcane cultivars.”
Revision of table 2 by adding the source of variation error and by adding the term “two-way” in the table heading in line 322: “Table 2. Two-way variance analysis (ANOVA) for the number apterous adult non-viruliferous Melanaphis sacchari across eight sugarcane cultivars in a free-choice bioassay.”
Revision of figure 3 by adding the means across timepoints obtained in the one-way ANOVA in lines 325-332: “Figure 3. A) Ln (x+5) transformed mean number of apterous adult non-viruliferous Melanaphis sacchari per sugarcane cultivar at short (0.5, 1, 3, 6 h) and long-term (24 h) intervals after the release of fifty individuals in a free-choice bioassay. The differences among cultivars are not significantly different according to the LSD test after one-way ANOVA for each timepoint (P > 0.05). Error bars represent standard error (n = 16). B) Ln (x+5) transformed mean number of adult non-viruliferous M. sacchari per sugarcane cultivar, each consisting of the mean of the five timepoints. Means with the same letter are not significantly different according to the LSD test after two-way ANOVA with all observations (P < 0.05). Error bars represent standard error (n = 80).”
Revision of the Results section based on the revised ANOVA in lines 307-317: “The one-way ANOVA for each timepoint of the free-choice bioassay revealed non-significant differences among cultivars for number of M. sacchari (p>0.05) (Supplementary table S7). The mean number of aphids across timepoints indicated increases from 0.5 to 3 h for the cultivars IACSP01-5503, IACSP96-7569, IACBIO-266, IACSP01-3127, and IACSP95-5000; decreases from 3 to 24 h for IACSP01-5503, IACSP96-7569, and IACSP95-5000; and increases from 3 to 24 h for IACSP01-3127. Another noteworthy change was the increase in the mean number of aphids from 6 to 24 h observed for IACCTC05-2562 (Figure 3a). None-theless, the two-way ANOVA with all 640 observations of the free-choice bioassay showed non-significant effects for time after release and its interaction with cultivar, while a significant F value (p<0.05) was observed for the variable cultivar (Table 2).”
Revision of the Discussion section addressing the biological implications of the findings of the free-choice bioassay in lines 385-398: “The results obtained in the free-choice bioassay revealed IACSP96-7569 as the least preferred cultivar, followed by IACSP95-5094, IACSP01-5503, and IACBIO266, which supports the expression of antixenosis among the cultivars evaluated here. Also, the apparent movement of aphids among cultivars throughout the time intervals of 0.5, 1, 3, 6, and 24 h after release further indicates that host factors may have affected the settling behavior of the non-viruliferous aphids. Such outcome differs from the reported by Akbar et al. [74], in which antixenosis assays revealed no preference of M. sacchari and Sipha flava among sugarcane cultivars, with aphids finding their hosts within an hour followed by no apparent movement among cultivars for the 24 h duration of the assay. Conversely, significant differences on aphid settling on potted sugarcane plantlets were observed by Fartek et al. [51] when comparing the cultivars R 365 and MQ 76/53 under no-choice conditions at 8, 24, 48, and 72 h after aphid deposit, indicating the expression of antixenosis in the former cultivar from 8 to 24 h, followed by a similar decrease in aphid populations between cultivars from 24 to 72 h.”
